# Electric Motor Vibration Signal Classification Using Wigner–Ville Distribution for Fault Diagnosis

**DOI:** 10.3390/s25041196

**Published:** 2025-02-15

**Authors:** Jian-Da Wu, Wen-Jun Luo, Kai-Chao Yao

**Affiliations:** 1Graduate Institute of Vehicle Engineering, National Changhua University of Education, Changhua 50007, Taiwan; 2Department of Industrial Education and Technology, National Changhua University of Education, Changhua 50007, Taiwan; wenjun_2020@ctu.edu.tw (W.-J.L.); kcyao@cc.ncue.edu.tw (K.-C.Y.)

**Keywords:** Wigner–Ville distribution method, brushless motor fault diagnosis, object detection, YOLO

## Abstract

Noise and vibration signal classification can be applied to fault diagnosis in mechanical and electronic systems such as electric vehicles. Traditional signal classification technology uses signal time and frequency domain characteristics as the identification basis. This study proposes a technique for visualizing sound signals using the Wigner–Ville distribution (WVD) method to extract vibration signal characteristics and artificial neural networks as the signal classification basis. A brushless motor is used as the machinery power source to verify the feasibility of this method to classify different signal vibration characteristics. In this experimental work, six states in various brushless motor revolutions were deliberately designed for measuring vibration signals. The brushless motor vibration signal is imaged using the WVD analysis method to extract the vibration signal characteristics. Through the WVD method, the brushless motor data is converted, and the YOLO (you only look once) deep coiling machine neural method is used to identify and classify the brushless motor WVD images. The Wagener analysis method parameters and recognition rates are discussed, thereby improving accurate motor fault diagnostic capabilities. This research provides a method for fault diagnosis that can be accurately performed without dismantling the brushless motor. The proposed approach can improve the reliability and stability of brushless motor applications.

## 1. Introduction

In traditional motor fault diagnosis, mechanical structures are inspected under operating conditions to identify and confirm issues. With advances in technology, sensors are now used to analyze mechanical faults like those in motors, enabling faster resolution. Vibration and sound data are commonly used to pinpoint fault locations. Accelerometers are often employed to collect vibration signals, recording device vibrations accurately. For vibration signal analysis, methods like Fourier transform (FFT) and wavelet transform are commonly used to extract signal features. However, FFT has limited resolution because it does not synchronize the time and frequency domains, leading to a loss of time-domain information when converting to the frequency domain. As a result, time-domain vibration signals cannot be fully observed.

This research uses the Wigner–Ville distribution to convert the brushless motor vibration signal into an image. YOLO object detection to identify fault types is then applied. Lohmann, A. W. [1] in 1993 defined the Wigner function distribution and described the Wigner and Fourier transformations from a geometric perspective. Meng, Q. and Qu, L. [2] in 1991 suggested using the Wigner distribution for rotating machinery fault diagnosis, showing that it can effectively analyze time-frequency characteristics and provide useful insights. Weinbub, J. and Ferry, D. K. [3] proposed the Wigner function in 1932, which offers important quantum method information relevant to physical systems.

In 2003, Geng, Z. et al. [4] proposed engine vibration analysis and diagnostic methods focusing on feature extraction from reciprocating engine vibrations. Zhang, Z. [5] in 2019 introduced the optimal linear normal Wigner distribution for noise linear frequency modulation signals, utilizing a three-parameter linear integral transformation. Brajović, M. and Popović-Bugarin, V. [6] in 2015 added the Wigner distribution algorithm to the ant optimization method to estimate instantaneous frequency in high-noise environments. Stankovic, L. and Katkovnik, V. [7] addressed challenges in using the Wigner distribution for time-frequency representation and proposed an adaptive algorithm to improve analysis. Qiu, L. [8] in 1993 introduced the Wigner–Ville distribution for analyzing non-stationary signals.

In 1999, Lee, S. K. [9] applied Wigner analysis to detect pulse signals in damaged gears, where such signals are often masked by noise. Dragoman, D. [10] in 2005 demonstrated the Wigner distribution’s versatility across various signal-processing fields, especially in optics. Chen, J. Y. and Li, B. Z. [11] developed short-term Wigner (STWD) and derived the WVD (Wigner–Ville distribution) as a tool for analyzing non-stationary signals, revealing multiple frequency components simultaneously. Liu, X. et al. [12] proposed optical applications, Wigner analysis enhances directional and local image features. Wu, J. D. and Huang, C. K. [13] proposed engine fault diagnosis based on intake manifold pressure analysis using Weigl’s method and highlighted the Wigner distribution’s ability to provide clear time-domain and frequency-domain energy spectra.

Huang, H. B. et al. [14] applied the Wigner distribution to analyze vehicle suspension noise and correlated their findings with expert evaluations, establishing an analysis method for vehicle chassis systems. Tang, B. et al. [15] used wavelet transformation and Wigner analysis for diagnosing large wind turbine faults. Tao, X et al. [16] proposed a bearing fault detection method based on wavelet transform and generalized Gaussian distribution modeling, which can use wavelet transform to extract vibration signal features. Dhandapani, R. et al. [17] proposed a bearing fault diagnosis and classification method based on generalized Gaussian distribution and multi-scale dispersed entropy features in 2022 and used machine learning for diagnosis. In deep learning, Lee, K. B. and Shin, H. S. [18] used algorithms to detect tunnel accidents, identifying reverse vehicles, parked cars, people, and flames within 10 s. Liu, J. [19] summarized deep learning theories in self-driving car image recognition. Widjojo, D. et al. [20] classified automobile sheet metal damage using CNN (Convolutional neural networks) for faster data classification. Rao, A.S et al. [21] in 2022 classified vehicle models using deep learning. Kulkarni, R. et al. [22] highlighted challenges in traffic light and lane detection for autonomous vehicles. Silva, J. D. et al. [23] applied micro machine learning to analyze potholes, enhancing model training by correcting image boundaries first. In agriculture, Kurniadi, F. A. et al. [24] used drones to photograph dairy cows and applied YOLO for effective cattle identification, noting reduced accuracy with increased flight altitude. Reddy, B. et al. [25] applied YOLO to identify motorcycle violations such as helmet-less riding or overloading. Sun, Z. [26] YOLO has also been applied for real-time airport safety monitoring. Liu, L et al. [27] published a method for estimating the remaining useful life (RUL) of lithium-ion batteries using small sample data. The Deep Autoregressive Recurrent Neural Network (DAR-RNN) proposed by Zhang, C. et al. [28] predicts the remaining useful life (RUL) of lithium-ion batteries throughout their entire life cycle. In order to enhance the classification capability of brushless motor faults, this study established a brushless motor fault vibration signal database. By applying the Wigner analysis method to these signals, the vibration signal characteristics are expressed, the Wigner analysis method parameters are further studied, and the analysis parameters are optimized to improve fault classification accuracy and reliability.

In order to enhance the brushless motor fault classification capability, this study established a brushless motor fault vibration signal database. By applying the Wigner analysis method to these signals, the vibration signal characteristics are expressed, and the Wigner analysis method parameters are further studied. The analysis parameters are optimized to improve the fault classification accuracy and reliability. This study applies YOLO object detection to Wigner–Ville for brushless motor fault diagnosis. First, we decided to conduct vibration analysis on brushless motors and use accelerometers to collect data on the fault conditions of brushless motors. The data was divided into six states. After establishing a brushless motor vibration database, we used Wigner–Ville to analyze the vibration data. Visualization is used as the basis for image recognition. In the image recognition software application, YOLO is selected for data testing, training and model establishment. The time delay parameters in Wigner–Ville are adjusted to find the most suitable method for analyzing the brushless motor vibration signal coefficient. The reason why we chose YOLO in the recognition system is that it performs well in image classification and improves its feature extraction capability in the architecture, which can maintain high image analysis accuracy. Therefore, we chose to use the YOLO system in image recognition as the main framework for feature recognition.

## 2. Feature Extraction Methods Using Wigner–Ville Distribution

The Wigner–Ville distribution is a time-frequency analysis method used to analyze the instantaneous signal frequencies. In this study the Wigner–Ville distribution is used to analyze brushless motor fault data, and the brushless motor working conditions are divided into six states. The following is the Wigner–Ville distribution derivation process. The Wigner–Ville distribution was proposed by the Hungarian physicist Eugene Wigner in 1932. The time-frequency mathematical form of the Wigner–Ville distribution is as follows: Equations (1) and (2).(1)$$Kx(t,\omega )=\int_{-\infty }^{\infty }x\left(t+\frac{\tau }{2}\right)\cdot x*\left(t-\frac{\tau }{2}\right)\cdot e^{-j\omega \tau }d\tau$$
$Kx\left(t,\omega \right)$ represents a Wigner–Ville distribution time-frequency function, showing the brushless motor vibration signal energy distribution across time and frequency.x(t) represents the brushless motor vibration signal, which is awaiting analysis.x∗(t) represents the conjugate of x(t). For a real-valued brushless motor vibration signal, the conjugate is equal to the signal itself. For complex signals, the conjugate means taking the negative of the imaginary part of the complex number.τ is the time delay coefficient, representing the difference between two time points of the brushless motor vibration signal.ω represents the brushless motor vibration signal frequency.j is the imaginary unit.$e^{-j\omega \tau }$ is the complex exponential function that describes the brushless motor vibration signal phase changes at different times and frequencies.dτ is a small increment of the time delay coefficient.

If Equation (1) is applied to a signal, Equation (1) becomes Equation (2).(2)$$Wx\left(t,\omega \right)=\int_{-\infty }^{\infty }x\left(t+\frac{\tau }{2}\right)\cdot x*\left(t-\frac{\tau }{2}\right)\cdot e^{-j\omega \tau }d\tau$$Equation (2) illustrates the brushless motor vibration signal energy distribution in time and frequency. The brushless motor vibration signal characteristics at different times can be observed.

The Wigner–Ville distribution principle can be understood from Equations (1) and (2). After these experiments it was found that the Wigner–Ville distribution can provide good time-frequency resolution in analyzing brushless motor vibration signals and determine various categories. The brushless motor vibration signal energy distribution is different, and the brushless motor vibration signal can be presented through the Wigner–Ville distribution two-dimensional image. When the brushless motor vibration signal changes in different states, the changes can be distinguished through the two-dimensional image, allowing the brushless motor vibration signal to be accurately identified. The brushless motor vibration signal is visualized to facilitate judgment of the brushless motor’s abnormal status, allowing the signal to be cross analyzed.

When the brushless motor vibration signal is converted from the signal acquirer through MATLAB 2023 (MathWorks, Natick, MA, USA), and before using Wigner–Ville on the signal, the Hilbert transformation is first performed on the signal. Time-frequency analysis is then performed on the brushless motor vibration signal. Hilbert transformation can be used to ensure better results for the signal. The Hilbert transformation can be understood from Equation (3).(3)$$H\{x(t)\}=\frac{1}{\pi }\int_{-\infty }^{\infty }\frac{x(\tau )}{t-\tau }d\tau$$
x(t) represents the input brushless motor vibration signal.t represents the brushless motor vibration signal time length.τ represents the Hilbert transformation time delay coefficient.


In signal processing a real signal number is converted into a complex signal number through Hilbert transformation. The original signal phase characteristics are retained after the analysis is completed. This conversion can be used to obtain more information from the brushless motor vibration signal. The brushless motor’s vibration characteristics can be extracted using Wigner–Ville. Hilbert transformation can capture the vibration signal energy changes in nonlinear systems and is suitable for use in analyzing mechanical vibrations.

In Figure 1 and Figure 2, the sine wave as the input signal and the brushless motor vibration signal are represented, with the characteristics displayed in 2D. The signal characteristics are shown on a plane, further highlighting that the brushless motor normal-state vibration signal characteristic values are well presented after Wigner–Ville and transformation.

## 3. Object Detection Principle

### 3.1. Principles of Deep Learning

Deep learning is a subset of machine learning and has become increasingly popular in recent years. One well-known deep learning algorithm is the convolutional neural network, which will be discussed in later chapters. Deep learning is a key part of artificial intelligence, built on machine learning principles. Machine learning uses large datasets and information from big data to analyze and solve problems, such as classification and identification. Historically, machine learning has been categorized into three types: supervised learning, unsupervised learning, and reinforcement learning, each contributing to various studies before deep learning was fully developed.

Supervised learning involves labeling the training data, defining its characteristics, and using it for tasks like regression, classification, and prediction. The computer is provided with the correct answers in advance, allowing it to train and correct the data to achieve better accuracy. However, supervised learning requires extensive preparatory work, and handling large datasets can be computationally challenging. Unsupervised learning allows the model to discover patterns in the data without predefined labels, helping the computer identify hidden rules. It is often used in cluster analysis and recommendation systems. However, without a standard answer, data distortion can occur. Reinforcement learning doesn’t require labeled data. Instead, the model learns through continuous interaction with its environment, optimizing its performance over time based on weighted outcomes. Figure 3 shows the evolution of deep learning.

Deep learning advances on these foundations using neural networks, which mimic biological brains. Initially, neural networks faced issues like vanishing gradients, but significant progress, especially since 2006, has led to the rise of multi-layer neural networks, driving the deep learning revolution. The development of neural networks began with the simulation of neural processes. However, challenges like vanishing gradients and complex calculations led to a period of stagnation. In 2006, Hinton et al. successfully trained multi-layer neural networks, marking a breakthrough in deep learning [29]. Deep learning, which allows neural networks to pass through multiple layers, initially struggled due to limitations in CPU computing power. However, with the advent of GPUs, deep learning experienced a surge in popularity. The neural networks concept, first introduced by Rosenblatt in 1958 as a single-layer model, evolved over time as scholars adopted multi-layer architectures, paving the way for modern deep learning [30].

### 3.2. Principles of Convolutional Neural Networks

Convolutional neural networks (CNN) analyze images by comparing local features, making them effective for image recognition even when the image is flipped or deformed. The convolution operation uses nonlinear transformation to extract features from images, but handling a large number of images can be computationally intensive. CNN reduces the dimensionality of images through pooling layers, which results in a feature map. Figure 4 Shows that the CNN structure includes an input layer, convolution layer, pooling layer, and fully connected layer. The input layer represents the image data.

The convolution layer acts as a filter, comparing local image features and calculating pixel correlations. Multiple layers can enhance the image features. The pooling layer compresses the image by retaining core features and reducing irrelevant data, which helps lower the computational load. The fully connected layer converts the 3D image data into 1D and functions as a classifier, enhancing image recognition by integrating features from previous layers. Figure 4 shows how the convolution layer works, applying filters to sharpen or blur the image by combining pixel values.

The pooling operation, shown in Figure 4, extracts maximum or average values from the feature map, reducing parameters while retaining important features. Pooling simplifies the image while preserving essential information for identification. The fully connected layer is illustrated in Figure 4. This layer integrates feature values from previous layers into one, improving image classification accuracy by reducing the influence of feature positions.

### 3.3. YOLO (You Only Look Once) Principle

Traditional object detection methods using classifiers fall behind the YOLO system in terms of speed and accuracy. YOLO, introduced by Joseph Redmon in 2015 with YOLO v1, has evolved to YOLOv8, improving both speed and applications. Unlike earlier methods, YOLO processes images by resizing them, running a single convolutional network, and setting thresholds for accuracy. By marking boundary coordinates and categories, it enables multi-object detection and improves real-time performance through neural network-based regression testing [31].

In YOLO, the input image is divided into cells, each responsible for detecting objects. Each cell predicts a bounding box with center coordinates (x, y), width (w), height (h), and a confidence score. The confidence score indicates the likelihood of an object being present in the cell and defines how confidence is calculated in YOLO’s end-to-end training which can be expressed as system $Pr(Object)*{IOU}_{pred\;}^{truth}$.

Binary values (0 and 1) can be used to determine the presence of an object in the cell, and the ${IOU}_{pred}^{truth}$. The value mentioned in $Pr(Object)*{IOU}_{pred\;}^{truth}$ represents the intersection of the detected object area in the image and the actual object area. This value is divided by the union of these two areas, as expressed in Equation (4). The model accuracy can be assessed based on the degree of overlap between the predicted object area and the actual object area. This assessment is particularly crucial when multiple objects are present in the image. Generally, a higher IOU value indicates a more accurate model prediction. A threshold is typically established, and lowering the IOU threshold during testing can result in a gap in accuracy. Usually, the IOU is set to be greater than 0.5.(4)$$IOU=\frac{detected\;object\;area\;and\;the\;actual\;object\;area\;intersection}{detected\;object\;area\;and\;the\;actual\;object\;area\;union}$$

In Figure 5, the molecular part illustrates the intersection between the object area detected by the two graphics and the actual object area, with the slashed portion representing this intersection. The denominator part depicts the detected and actual object areas, while the slashed part represents their union. Figure 5 is a schematic diagram of the IOU.

In the YOLO algorithm, accuracy is a crucial parameter, defined as the proportion of correctly detected results. It includes the ratio of correct predictions to prediction errors. This can be assessed using a confusion matrix, which categorizes results into four types: true positives (TP), true negatives (TN), false positives (FP), and false negatives (FN), as shown in Table 1.

For instance, in the WVD transformation image context, TP indicates correct identification of a normal brushless motor state, while TN reflects the correct identification of a fault. FP occurs when a normal state is misidentified as a fault, and FN is when a faulty image is incorrectly classified as normal. The system accuracy can be calculated using the Equation for Precision in Equation (5) which relates TP and FP to the total number of correctly identified samples(5)$$Precision=\frac{TP}{TP+FP}$$

There is also a very important numerical recall rate in the YOLO algorithm. The recall rate refers to how many numbers in the TP are correctly predicted in the original sample. I In the recall rate, it will be observed that the correct number of false negatives (FN) identifying misidentifications can be used to test the system’s sensitivity. When the recall rate is low, it means that samples that should have been identified will be missed during prediction, so the system recall rate is expected to reach 100%, that is, the larger the recall rate value, the better, which means that the model can avoid omissions when predicting, and the recall rate Equation is as shown in Equation (6).(6)$$\;\;\;\;Recall=\frac{TP}{TP+FN}$$

Mean average precision (mAP) is the average accuracy in multiple types of problems. Average precision (AP) is the average accuracy of different categories. mAP is an indicator commonly used to measure performance in machine learning models and can evaluate the model as shown in Equation (7). Whether it is effective, when identifying the brushless motor WVD type figures, mAP is also used as an indicator, and the accuracy of all categories can be obtained to determine whether various brushless motor WVD type figures can be effectively identified. To facilitate species identification, the following equation is used:(7)$$mAP=\frac{1}{n}\sum_{k=1}^{k=n}AP_{k}$$

Among them, $AP_{k}$ means that there are k types of AP in the commonly recognized figures, and n = is the number of types.

YOLO (You Only Look Once) has diverse applications, functioning as both a figure recognition and a real-time image recognition system. It represents an innovative approach in image recognition, significantly enhancing detection speed. In contrast to previous methods like R-CNN (Regions with Convolutional Neural Networks), which are time-consuming and computationally intensive, YOLO streamlines the process.

YOLO operates by grid-dividing the image beforehand to facilitate boundary selection for identifying multiple objects within a single figure. The model then trains the image through segmentation and prediction using a deep convolutional neural network with forward propagation. The YOLO system has evolved through several versions. YOLOv1, the original version, markedly improved detection speed but still had accuracy issues. YOLOv2, released in 2017, introduced batch normalization to mitigate the gradient disappearance, enhance training speed, and incorporate multi-scale training. YOLOv3 followed in 2018, featuring Darknet-53 with 53 convolutional layers and a Residual network to manage gradient issues, improving small object prediction. In 2020, YOLOv4 was launched, focusing on further speed and accuracy improvements over YOLOv3. The same year, YOLOv5 was introduced, leveraging the PyTorch framework for easier sample training while enhancing computational efficiency. YOLOv6, released in 2021, doubled the speed of its predecessor while maintaining accuracy. YOLOv7, announced in 2022, upgraded its backbone network and added advanced feature extraction techniques. Finally, YOLOv8, unveiled in 2023, combines the strengths of YOLOv7, incorporating new anchor-free detection and closed mosaic enhancement for versatile sample training. YOLOv8 was launched in 2023. It combines the features of the previous generation with sculptural improvements. The Anchor-Free design can predict the boundaries or key points of the object. It can perform the position and shape of the object through the central body or rectangle of the object. The Anchor-Free design can reduce design overhead and reduce the number of calculations, improve ability to learn, and avoid excessive problems.

## 4. Discussion

### 4.1. System Structure

This study used brushless motor vibration signals with different types of faults to identify fault types. The data acquisition equipment includes an accelerometer, a data acquisition card NI-6024E (National Instruments, Austin, TX, USA), and a data acquisition module NI-9233 (National Instruments, Austin, TX, USA). The accelerometer is installed on the brushless motor with an accelerometer and data acquisition system to record vibration signals. The brushless motor vibration signal is acquired through LabVIEW (National Instruments, Austin, TX, USA), the sampling frequency is set to 1000 Hz, the sampling time is 3 s, with 3000 points, and the number of samples for each fault type is 50 pieces of data, with a total of six settings projected, and a total of 300 data. During the training process, the 300 data were split into 240 data as training data and 60 data as verification data for model training.

A fault test was conducted on the brushless motor equipment. Because the brushless motor has a wide range of applications and the stability of its surrounding systems are also taken into consideration, six related fault items were set, namely loose screws on the brushless motor bracket. The brushless motor bracket screws are loose, the brushless motor bearings are worn, the brushless motor body fixing screws, the brushless motor shaft center is worn, and the brushless motor is in a normal state. The fault categories and sample numbers are in Table 2.

Basic brushless DC motor control, compared with traditional brushed motors, requires driver control to realize its effect. By receiving signals from the controller to confirm the rotor position, feasible control signals can be sent to adjust the DC motor. Brushless motors can achieve electrical commutation using power components to replace the traditional brushed motor carbon fibers. Brushless motors are relatively simple to maintain and have a long life.

The brushless motor used in this paper is X4120II KV400 brushless motor (Yuan-hang Technology, China), which is often used in drones. For the brushless motor characteristics discussion, this motor is used as the experimental framework because brushless motors are very important in life. Brushless motors are used in everything from daily home appliances to mobile vehicles. The difference between brushless motors and traditional brushed motors is that they can switch the current direction through electronic commutation. Compared with traditional brushed motors, they have the advantage of high efficiency and can provide the same power. In terms of size, it can be made smaller, which is an advantage for use in unmanned vehicles. This paper next discusses brushless motor maintenance compared to traditional brushed motors. Because of component wear such as carbon brushes, brushless motors can have simpler maintenance. In the past, control accuracy was a disadvantage in brushless motors, but with the current technological progress and improvement in controllers, the control circuit can be more precise and use higher frequencies for control, giving brushless motors more accurate control. This paper decided to study and explore the faults and states of brushless motors, trying to find a way to easily identify the brushless motor fault states and predict faults in advance to protect equipment and user safety. In the experimental process the brushless motor vibration signals were plotted using the Wigner analysis method.

The brushless motor model used in this paper is X4120II KV400, where KV400 means that each volt can produce a speed of 400 RPM. In this paper, 2000 RPM is used for experimental sampling. The motor stator outer diameter is 41 mm, the number of stator slots is 24N, the number of rotor poles is 22P, the rotor diameter is 46.2 mm, and the length of the brushless motor is 31.5 m. Detailed specifications are shown in Table 3.

The equipment used in the experiment is built on the Windows 11 environment, and the training process uses the CPU and GPU respectively, as mentioned in Table 4. Deep learning operating environment and equipment specifications. The brushless motor Wigner characteristic diagram is drawn using MATLAB 2023 (MathWorks, Natick, MA, USA), which can present the characteristics of various types of brushless motor fault diagrams in the form of figures.

### 4.2. Experimental Works and Data Measurement

This study used the WVD to study brushless motor vibration signal fault diagnosis and extends the WVD sampling point study to find the Wagner analysis applicable parameters for the brushless motor vibration signals, among which the number of WVD sampling points were studied individually. The sampling points are N = 2, 5, 10, 15, 20, 25, 30, showing the brushless motor vibration signal diagram in each parameter and the experimental process. First of all, this research simulated brushless motor states and built an experimental platform for signal analysis. As shown in the brushless motor experimental architecture diagram in Figure 6. the motor experimental combination was constructed into the experimental platform. The platform flatness was confirmed. The brushless motor body is fixed onto the brushless motor bracket. The brushless motor bracket was set up on the experimental platform. The brushless motor stability with the connected platform and the relationship between the brushless motor and connected platform was studied in this experiment. When used in equipment, the fault state is detected and the WVD is used to monitor the various brushless motor states and identify the most suitable experimental parameters for the brushless motor. 

Brushless DC Motor Fault Diagnosis Flowchart. It is first decided to use a brushless DC motor to simulate the fault state in the experiment, and then the vibration accelerometer is installed. After trying each position, the vibration is accelerated. The gauge is installed in the appropriate position, and the data collection is completed through the data acquisition system (NI-6024E) and data acquisition card (NI-9233). Through MATLAB 2023 (MathWorks, Natick, MA, USA) the brushless motor vibration signal is collected using WVD and the analysis method was used to draw the figure, and the WVD parameters were adjusted. We tried to find the parameters suitable for use on the brushless motor and successfully used YOLO v8 to draw the brushless motor WVD chart. In the past, brushless motor fault diagnosis and their platforms required disassembly to clearly determine which side had a fault. In this study the brushless motor and the surrounding structure fault vibration signals are collected. This allows for a faster and more convenient way to diagnose faults in electric vehicles. Figure 7 is the time-frequency diagram and Fourier transform image of the brushless motor in a normal state. It can be seen that the signal is a low-frequency signal, and the extraction length is 3 s. After Fourier transform, obvious features can be seen. The research process of this study is as shown in Figure 8. Brushless DC Motor Fault Diagnosis Flowchart.

Brushless motor Wagner figure with N (Scaling factor) 2~30. The changes in the brushless motor vibration signal after WVD analysis can be observed in Figure 9. When the brushless motor is in a normal state, the frequency distribution is relatively regular. Under fault conditions, respective abnormal characteristics will appear, which are key indicators for identifying the fault condition of the brushless motor. When N (the scaling factor) gradually increases, more cross-interference can be observed in the image. The impact of N (the scaling factor) on image recognition performance is explained in Table 5 and Table 6.

### 4.3. Object Detection Based on Wigner–Ville Distribution Motor Vibration Signal Fault Diagnosis Analysis

In this experiment, the vibration signal was analyzed for various brushless motor states. In this experiment the first parameter changed is the parameter N for the number of sampling points in the WVD analysis method. N is the scaling factor that affects the time scale. The WVD analysis method was successfully used to identify the brushless motor status and determine the parameter N used for the number of sampling points as the experimental change parameter. It can be seen that when the number of sampling points changes, the brushless motor WVD analysis diagram will change significantly. However, what needs to be confirmed in the experiment is what parameters are the most suitable parameters for analyzing the brushless motor vibration signals. This was verified in this experiment. In the YOLOv8 picture training the appropriate number of iterations is shown. In addition to adjusting the number of sampling points N in the WVD analysis method, various iterations are also applied when training the models. The goal is to find a smaller number of layers that still achieve a high mAP while confirming the WVD parameters. The experimental results are shown in Table 5. The calculation was carried out from iteration number 1 to 50 according to 1, 10, 20, 30, 40, 50. The results will be calculated and the data recorded in Table 5. It can be seen that when the parameter N = 2, the operation result is the best, and according to the operation result from the N = 2 part of the statistical results, the best result is presented in Table 6 when the number of iterations is 40.

The brushless motor vibration signals were successfully imaged using WVD. During the image formation process, the changes in the key parameter N (Scaling factor) in the WVD were discussed, mainly focusing on the feature extraction points.

The model can be used to accurately predict brushless motor faults based on WVD. Figure 10a shows the confusion matrix for brushless motor fault detection. The confusion matrix provides a clear view of the model’s prediction accuracy, indicating whether false positives or false negatives are present and displaying the proportion of these errors. The prediction accuracy for the brushless motor failure in the training model shows that the status of each type of brushless motor has been correctly judged and analyzed. In Figure 10b,c when the Scaling factor N = 2 and the number of iterations is set to 40, the F1-confidence curve and the recall-confidence curve demonstrate strong performance. The reason for using a scaling factor of N = 2 and 40 iterations is that, based on previous experimental data, this ratio and number of iterations yield the best results. Therefore, the F1 curve of the training results using this ratio shows that the model performs fault identification with high confidence.

When comparing research methods, this study also used the Pseudo Wigner–Ville Distribution (PWV) for feature extraction. Both perform quite well in terms of recognition results such as recognition rate. In the future, WVD and PWV can be applied to brushless faults application, because their effectiveness, reliability and practicability in fault diagnosis systems have been proven in this study. As shown in Table 7, when the PWV parameters are adjusted to 2 and the number of iterations is 50, the training data results for each brushless motor state were obtained.

## 5. Conclusions

This research identified new brushless motor analysis methods and used them as the experimental basis for higher frequencies currently used in the field of electric vehicles. Brushless motors are suitable for many fields, such as drones, electric vehicles, electric motorcycles, etc. This paper attempted to find a method that can identify brushless motor fault signals and, through graphical methods, present the motor vibration signal. After experimentation, this study chose the WVD to extract brushless motor vibration signal features. During these experiments, it was proven that the WVD is useful. For the brushless motor vibration signals, with good signal analysis, this research successfully used the brushless motor vibration signals to clearly determine the brushless motor fault characteristics using the WVD. The vibration fault signal can be effectively seen from the picture.

Neural-like methods have been used to identify images in the past. In recent years, with the booming development of artificial intelligence, YOLOv8 was chosen to classify brushless motor WVD images quickly and efficiently. Through image gallery training and testing, this research successfully completed brushless motor WVD vibration signal fault figure classification. This research further studied the WVD analysis method interval parameters and obtained the vibration signal suitable for use in brushless motors. The optimal interval parameter was identified using the mAP50-95 data affected by each parameter change. When the interval parameter exceeds 20, mAP50-95 will gradually decrease. Therefore, it is recommended that the brushless motor fault state be identified using the WVD interval parameter set to less than 20, so that it is easier to obtain significant identification effects. This study used WVD and YOLO to classify vibration signals for motor fault diagnosis. These algorithms can successfully and accurately determine what kind of fault state the brushless motor will be in. In the future, the proposed method will provide effective electric mobile vehicle fault diagnosis.

## Figures and Tables

**Figure 1 sensors-25-01196-f001:**
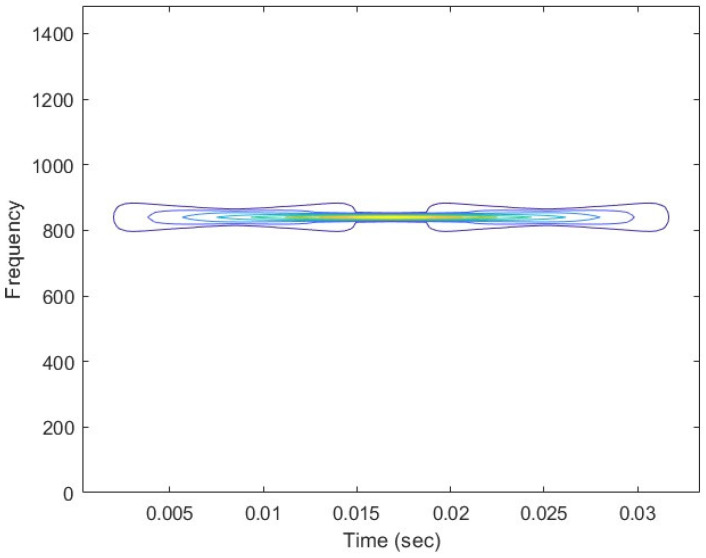
Sine wave WVD 2D time-frequency energy diagram. The bluer areas indicate lower vibration energy, while the green areas represent moderate vibration energy, and the yellow regions correspond to higher vibration energy.

**Figure 2 sensors-25-01196-f002:**
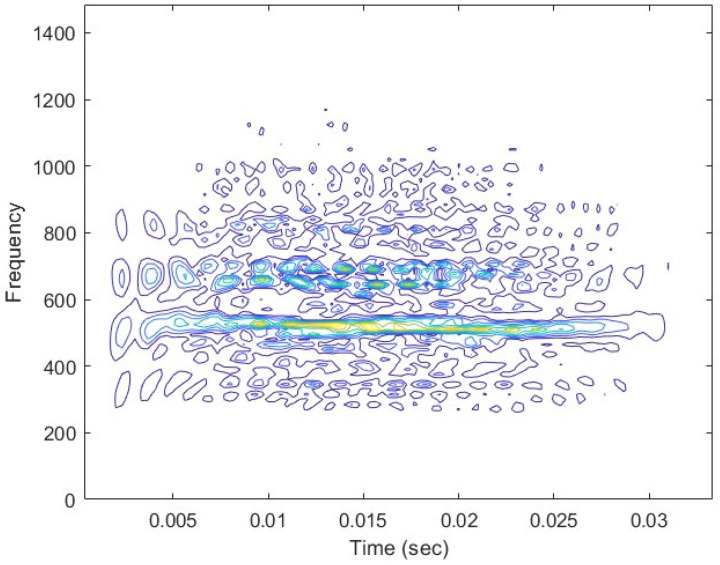
Brushless motor normal state vibration signal WVD 2D time-frequency energy diagram. The bluer areas indicate lower vibration energy, while the green areas represent moderate vibration energy, and the yellow regions correspond to higher vibration energy.

**Figure 3 sensors-25-01196-f003:**
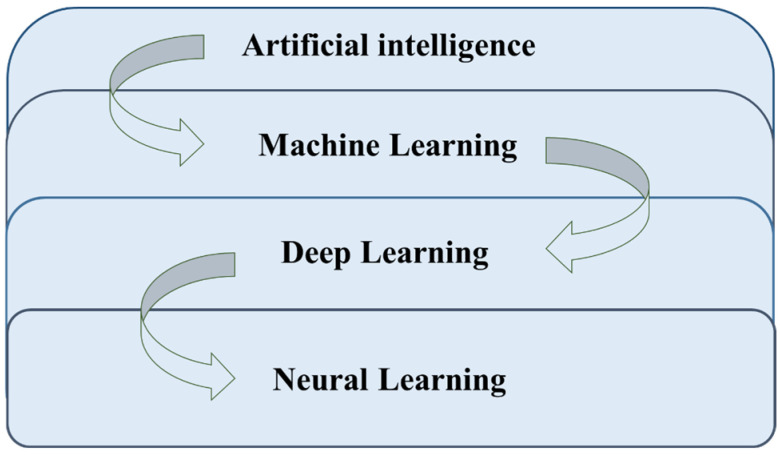
The evolution of deep learning.

**Figure 4 sensors-25-01196-f004:**
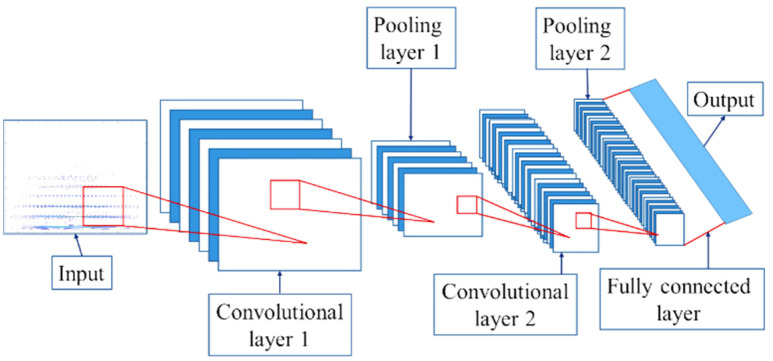
CNN architecture.

**Figure 5 sensors-25-01196-f005:**
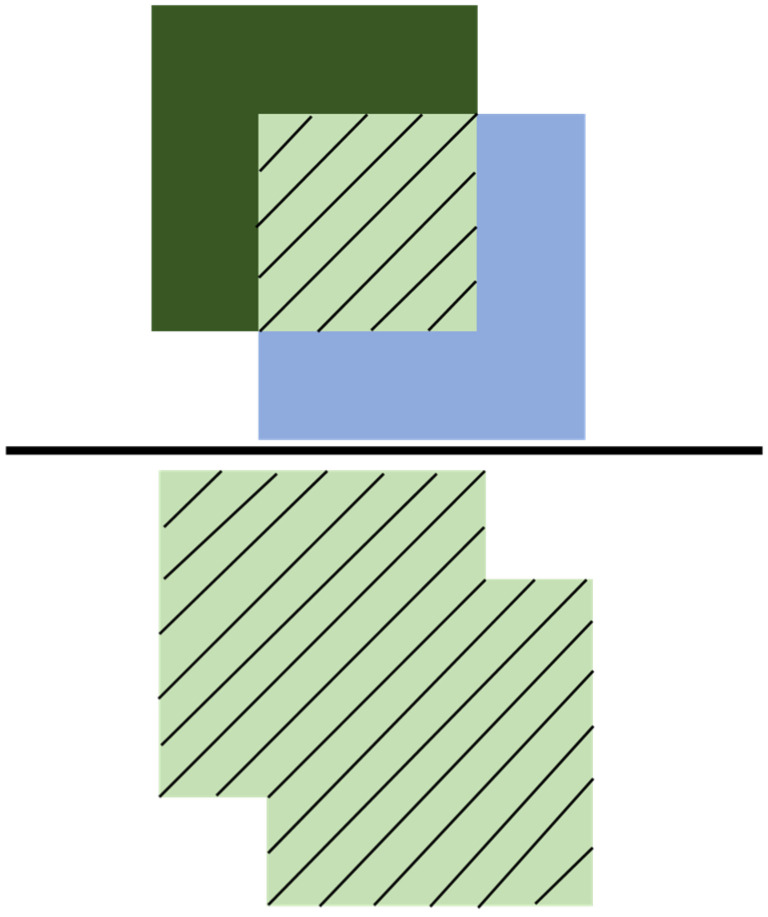
IOU diagram.

**Figure 6 sensors-25-01196-f006:**
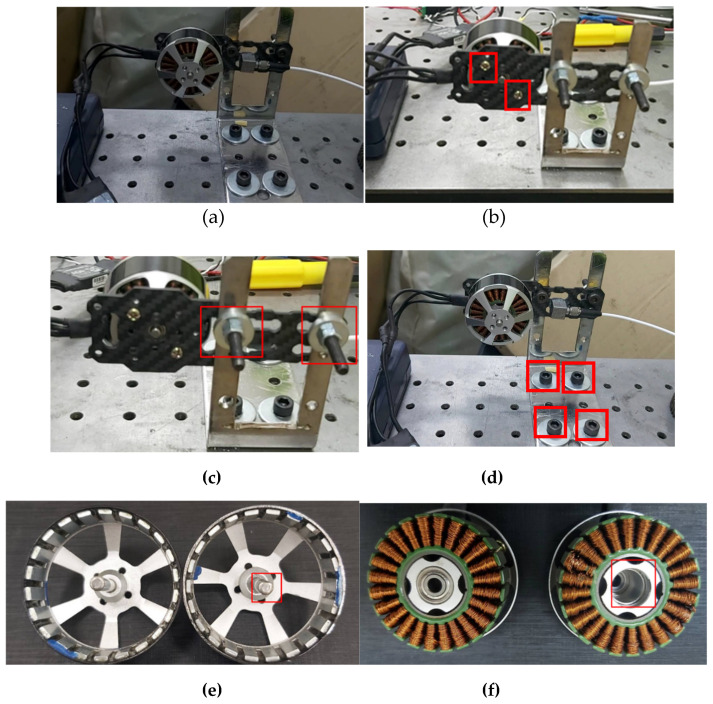
Brushless motor experimental architecture diagram. (**a**) Normal brushless motor status diagram; (**b**) Motor fixed screw loose, the fixing screws of the motor body are loose; (**c**) Motor bracket screw loose is the brushless motor bracket’s fault; (**d**) Motor bottom screw loose is the motor bracket base screw loose; (**e**) Motor shaft wear means the middle shaft of the motor is damaged; (**f**) Motor bearing not fixed means the motor bearing is damaged.

**Figure 7 sensors-25-01196-f007:**
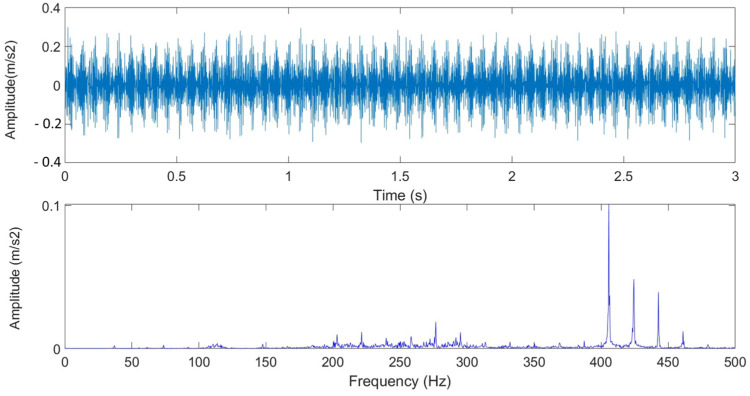
Time-frequency diagram and Fourier transform image of the brushless motor in normal state.

**Figure 8 sensors-25-01196-f008:**
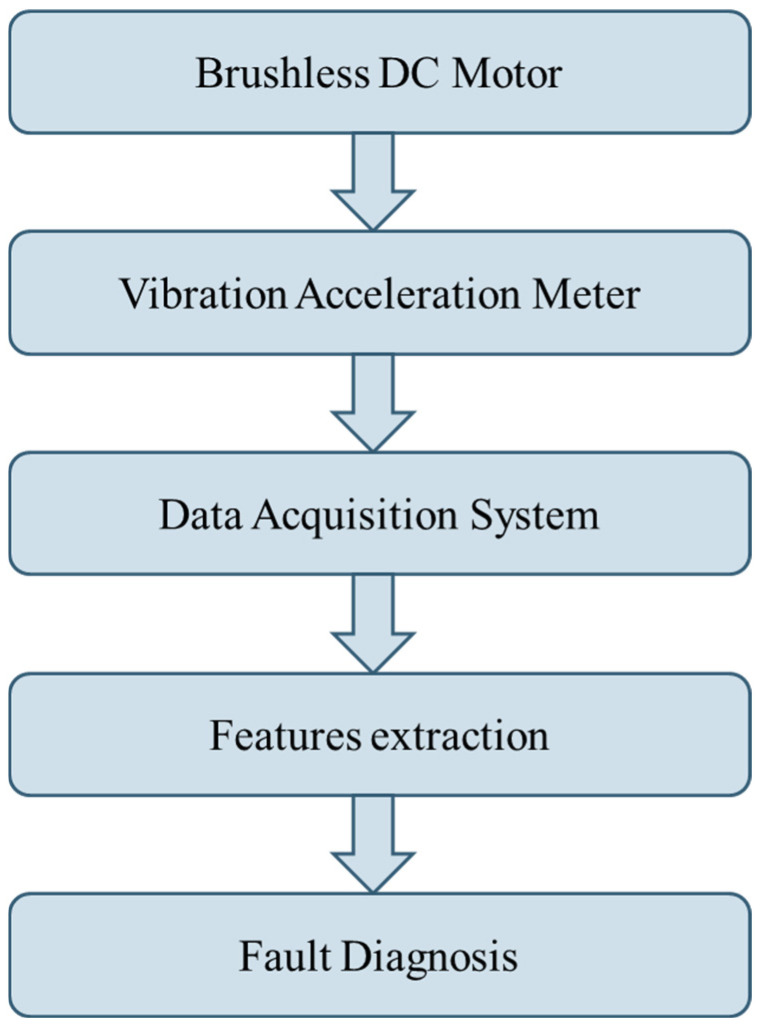
Brushless DC Motor Fault Diagnosis Flowchart.

**Figure 9 sensors-25-01196-f009:**
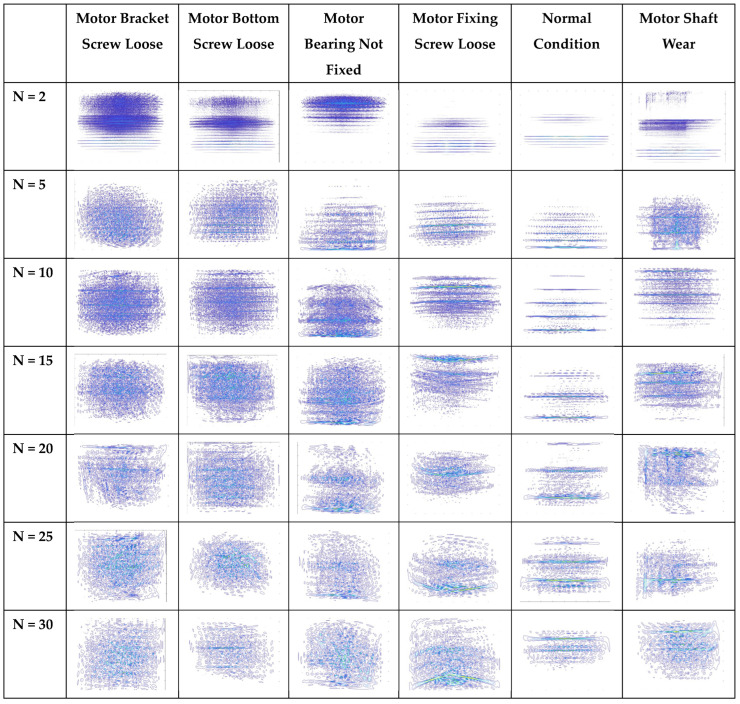
Brushless motor WVD figure with sampling points 2~30.

**Figure 10 sensors-25-01196-f010:**
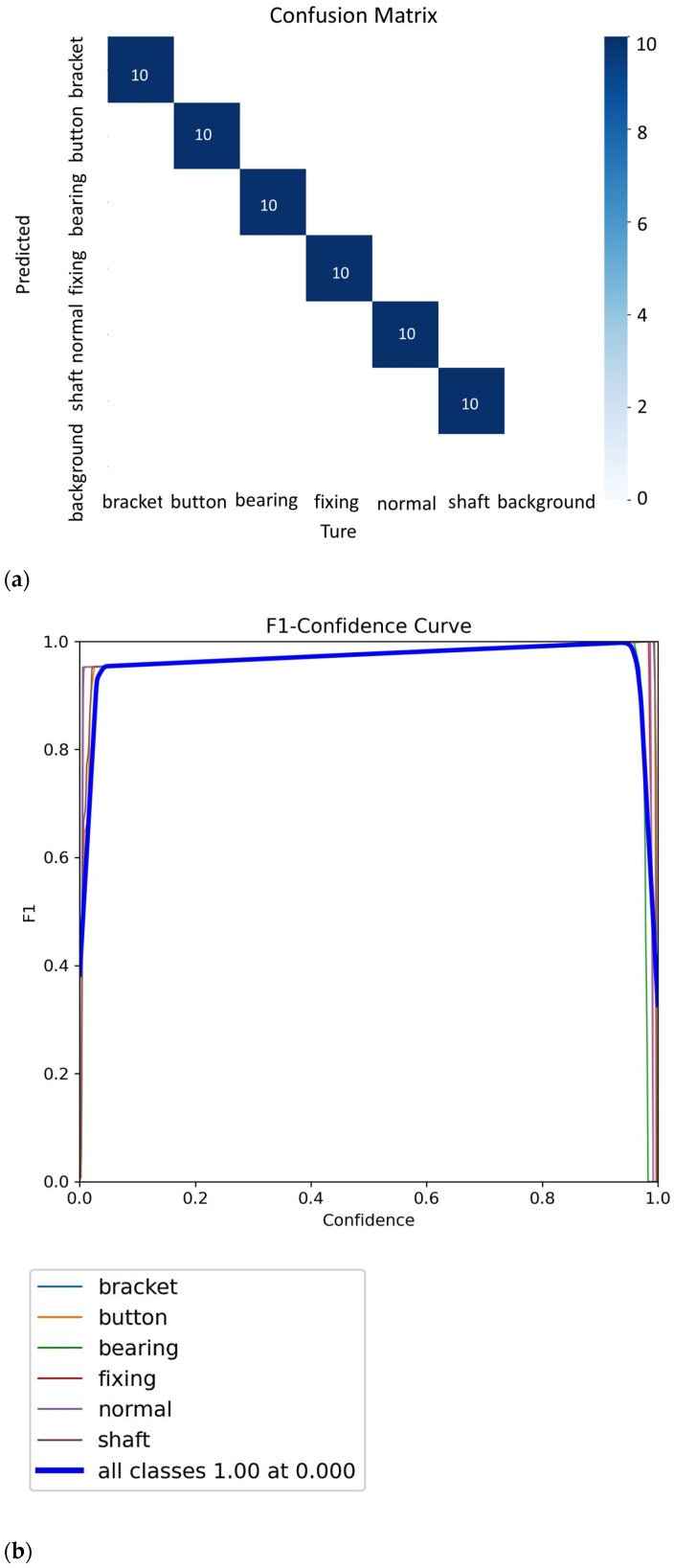
When the interval parameter N = 2 and the number of iterations is equal to 40, (**a**) confusion matrix figure; (**b**) confidence curve; (**c**) Recall-confidence curve figure.

**Table 1 sensors-25-01196-t001:** Confusion matrix table.

Confusion Matrix		Actual Values	
		Positive	Negative
Predicted	Positive	TP	FP
	Negative	FN	TN

**Table 2 sensors-25-01196-t002:** Brushless motor fault categories and sample number table.

Class	Samples
Normal	50
Motor fixing screw loose	50
Motor bracket screw loose	50
Motor bottom screw loose	50
Motor shaft wear	50
Motor bearing not fixed	50
Total	300

**Table 3 sensors-25-01196-t003:** Brushless motor specification sheet.

Brushless Motor Specification Sheet	
Specifications	X412011
Stator Diameter	41 mm
Stator Arms	24 N
Stator Poles	22 P
Motor KV	400
Rotor Diameter	46.2 mm
Motor Body Length	31.5 mm

**Table 4 sensors-25-01196-t004:** Deep learning operating environment and equipment specifications.

Equipment Specifications.	
Operating System	Windows 11
CPU	Intel(R) Core(TM) i7-10870H CPU
GPU	GeForce GTX 1650
RAM	32.0 GB
MATLAB	2023

**Table 7 sensors-25-01196-t007:** When the PWV parameters are adjusted to 2 and the number of iterations is 50, the training data results for each brushless motor state were obtained.

Class Name	P (%)	R (%)	mAP50 (%)	mAP50-95 (%)	F1
Motor bracket screw loose	99.4	100	99.5	99.5	99.7
Motor bottom screw loose	99.4	100	99.5	99.5	99.7
Motor bearing not fixed	99.4	100	99.5	99.5	99.7
Motor fixing screw loose	99.4	100	99.5	99.5	99.7
Normal condition	99.5	100	99.5	93.7	99.75
Motor shaft wear	99.5	100	99.5	99.5	99.75
all	99.4	100	99.5	99.5	99.7

**Table 5 sensors-25-01196-t005:** mAP50-95 statistics interval points.

N (Scaling Factor)	mAP50-95 (%)
2	85.25
5	87.28
10	84.57
15	83.05
20	75.87
25	71.87
30	66.42

**Table 6 sensors-25-01196-t006:** When the WVD parameters are adjusted to 2 and the number of iterations is 40, the training data results for each brushless motor state were obtained.

Class Name	P (%)	R (%)	mAP50 (%)	mAP50-95 (%)	F1
Motor bracket screw loose	99.3	100	99.5	99.5	99.65
Motor bottom screw loose	99.2	100	99.5	99.5	99.60
Motor bearing not fixed	99.6	100	99.5	99.5	99.80
Motor fixing screw loose	99.3	100	99.5	93.5	99.65
Normal condition	99.3	100	99.5	99.5	99.65
Motor shaft wear	99.3	100	99.5	99.5	99.65
all	99.3	100	99.5	98.5	99.67

## Data Availability

The original contributions presented in this study are included in the article. Further inquiries can be directed to the corresponding author(s).

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
