# Peer review of "Electric Motor Vibration Signal Classification Using Wigner–Ville Distribution for Fault Diagnosis"

_sensors, 2025, doi:10.3390/s25041196_

Round 1

Reviewer 1 Report

Comments and Suggestions for Authors

This research explores advanced analysis methods for brushless motors, focusing on fault signal detection and classification to enhance electric vehicle performance. It employs the Wegener analysis method to extract motor vibration signal features, proving its effectiveness in identifying motor faults through graphical representation. Additionally, YOLOv8, a neural-based AI method, was utilized for efficient classification of Wegener images. The study optimized interval parameters for fault diagnosis, recommending values under 20 for better accuracy. These methods successfully diagnose brushless motor faults and contribute to advancing fault diagnosis technology, supporting the electrification trend in mobile vehicles for improved reliability and performance.

There are question about the experiment. First, are the 300 samples sufficient to train the model? Since this is YOLO and deep learning usually require plenty of training data. Second, the testing dataset should be introduced.

Author Response

Revised paper of “Electric motor vibration signal classification using Wigner-Ville distribution for fault diagnosis”

Dear Editorial Board

Please find the attached file of the revised paper entitled “Electric motor vibration signal classification using Wigner-Ville distribution for fault diagnosis” which we submitted to “sensors” in December, 2024.  Modifications that have been made to this paper in response to the reviewer’s comments are summarized as follows:

Response to Review #1

Thank you for your valuable suggestions. Regarding the number of samples, since this study focuses on the specific fault detection and classification of a single detection project, the required number of samples is relatively lower than that of multi-classification or multi-target detection tasks.

This study selected YOLOv8 as the model, which has the ability to perform transfer learning on small data sets and can effectively improve the performance of the model. At the same time, the use of Wigner analysis method to analyze the vibration signal further enhances the recognition effect.

The paper has been considerably modified. Authors sincerely hope that these modifications have addressed all the points raised by the reviewers and the consistency of the technical content has been improved. Thank you and the referees very much for time and effort needed for the thorough review and the valuable comments on this paper.  I look forward to hearing from you soon.

With best regards,

Jian-Da Wu, Professor

Graduate Institute of Vehicle Engineering

National Changhua University of Education

1 Jin-De Rd., Changhua 500, Taiwan, ROC

Email: jdwu@cc.ncue.edu.tw

Reviewer 2 Report

Comments and Suggestions for Authors

With great interest, I’ve got acquainted with the research on the electric motor faults diagnosis with features obtained from distribution approximation. Authors have found that the Wigner-Ville distribution (WVD) can provide good time-frequency resolution in analyzing brushless motor vibration signals and thus be useful for analysis of different types of abnormal states. WVD is applied to the Hilbert transform of the analyzed signal. 2D WVD diagrams are then transferred to YOLO convolutional network (CNN) for final classification.

The paper contains an original experimental study, which increases its significance and contribution of the authors. High precision devices such as NI-6024E are used to record the data.

The paper is well written and illustrated. However, a number of shortcomings and comments may be listed.

1. Wegener and Wigner are mixed up. Alfred Wegener was not the mathematician (as Wigner was), and is famous for developing a robust hypothesis of continental drift, which is not in the scope of the article. Thus, the research is based on the use of Wigner-Ville distribution, which is also confirmed by multiple references to Wigner in the cited literature, and no one of Wegener. I recommend changing titles of the article and some subsections.

2. In referenced literature, authors focus only on Wigner distribution in electric motors diagnosis, though other types of distributions are applied for this task, see e.g. study on bearings fault detection using wavelet transform and generalized Gaussian density (GGD) modeling. I recommend extending the Introduction with a brief description of the application of statistical methods for diagnostics of electric motors.

3. In the dataset description, I recommend providing more information about the collected data: time plots, spectrograms, RQA analysis graphs, various metrics. It should be shown that the data requires developing new sophisticated approach for classification.

4. The proposed WVD-based feature extraction approach is not compared with other methods. For example, in recent research on time-series feature extraction by return map analysis to bearing-fault detection, the proposed feature extraction method is compared to GGD and entropy-based methods. I recommend making a similar comparison in your paper, including summary tables and plots.

5. I recommend adding F1-score to the list of calculated metrics.

6. It would be interesting to see the embeddings resulting from sequential layers of CNN

Author Response

Revised paper of “Electric motor vibration signal classification using Wigner-Ville distribution for fault diagnosis”

Dear Editorial Board

Please find the attached file of the revised paper entitled “Electric motor vibration signal classification using Wigner-Ville distribution for fault diagnosis” which we submitted to “sensors” in December, 2024.  Modifications that have been made to this paper in response to the reviewer’s comments are summarized as follows:

Response to Review #2

With great interest, I’ve got acquainted with the research on the electric motor faults diagnosis with features obtained from distribution approximation. Authors have found that the Wigner-Ville distribution (WVD) can provide good time-frequency resolution in analyzing brushless motor vibration signals and thus be useful for analysis of different types of abnormal states. WVD is applied to the Hilbert transform of the analyzed signal. 2D WVD diagrams are then transferred to YOLO convolutional network (CNN) for final classification.

The paper contains an original experimental study, which increases its significance and contribution of the authors. High precision devices such as NI-6024E are used to record the data.

The paper is well written and illustrated. However, a number of shortcomings and comments may be listed.

1.Wegener and Wigner are mixed up. Alfred Wegener was not the mathematician (as Wigner was), and is famous for developing a robust hypothesis of continental drift, which is not in the scope of the article. Thus, the research is based on the use of Wigner-Ville distribution, which is also confirmed by multiple references to Wigner in the cited literature, and no one of Wegener. I recommend changing titles of the article and some subsections.

Thank you for pointing this out. We have addressed the issue regarding the incorrect name entry and have made the necessary corrections. We appreciate your attention to this detail and your valuable feedback.

  1. In referenced literature, authors focus only on Wigner distribution in electric motors diagnosis, though other types of distributions are applied for this task, see e.g. study on bearings fault detection using wavelet transform and generalized Gaussian density (GGD) modeling. I recommend extending the Introduction with a brief description of the application of statistical methods for diagnostics of electric motors.

In response to this suggestion, the referenced literature in this study has already incorporated the relevant content, specifically in references 16 and 17.

Reference 16 (Tao et al., 2020) discusses the application of wavelet transform and generalized Gaussian density (GGD) modeling for fault detection, providing a robust statistical method to analyze and detect abnormalities in bearing systems.

Reference 17 (Dhandapani et al., 2022) extends this approach by combining generalized Gaussian distribution modeling with multiscale dispersion entropy features, further improving the diagnostic accuracy and classification precision of bearing faults.

  1. In the dataset description, I recommend providing more information about the collected data: time plots, spectrograms, RQA analysis graphs, various metrics. It should be shown that the data requires developing new sophisticated approach for classification.

Following your suggestion, the suggested content has been added to row 403 and the dataset description in Figure 7. It includes the reasons for establishing the database, as well as newly added pictures of the time domain and frequency domain of the vibration signal of the brushless motor under normal conditions, and provides explanations.

  1. The proposed WVD-based feature extraction approach is not compared with other methods. For example, in recent research on time-series feature extraction by return map analysis to bearing-fault detection, the proposed feature extraction method is compared to GGD and entropy-based methods. I recommend making a similar comparison in your paper, including summary tables and plots.

In line 465 and Table 7, there is a supplement about the method of Pseudo Wigner-Ville Distribution (PWV), and based on the conclusions obtained in the original WVD study, the experiment was started with parameter N. Subsequent The entire experiment will be improved based on this method to provide a feature extraction method more suitable for fault diagnosis of brushless motors.

  1. I recommend adding F1-score to the list of calculated metrics.

In accordance with your suggestion, the value of F1 has been added to Table 6 for reference.

  1. It would be interesting to see the embeddings resulting from sequential layers of CNN

Due to space constraints, not all iteration results are included in this article, but Table 6. shows the results when the number of iterations is 40.

The paper has been considerably modified. Authors sincerely hope that these modifications have addressed all the points raised by the reviewers and the consistency of the technical content has been improved. Thank you and the referees very much for time and effort needed for the thorough review and the valuable comments on this paper.  I look forward to hearing from you soon.

With best regards,

Jian-Da Wu, Professor

Graduate Institute of Vehicle Engineering

National Changhua University of Education

1 Jin-De Rd., Changhua 500, Taiwan, ROC

Email: jdwu@cc.ncue.edu.tw

Reviewer 3 Report

Comments and Suggestions for Authors

1. This paper proposes a technique for identifying motor fault vibration signals. Firstly, the Wegener method is used to extract vibration signal features, and then the YOLO algorithm is used for fault identification. However, the details of these technologies are not presented in the abstract, which cannot highlight the innovation of this study. Suggest presenting the specific innovative content proposed in this paper in the abstract.

2. In the introduction, the author cites a large number of references to verify the multifunctional role of Wegener analysis method in the field of signal processing and the application of YOLO algorithm in the field of computer vision. However, the author did not summarize the above research at the end of the introduction, pointing out the shortcomings that still exist in previous studies and the important contributions proposed in this paper. Suggest adding these contents before introducing the paper layout in the introduction.

3. Among the references cited by the author, whether it is Wegener or YOLO method, only a small number of references are relevant to the research objectives of this paper. Suggest the author to enrich the literature content from multiple perspectives, such as: An efficient diagnostic strategy for intermittent faults in electronic circuit systems by enhancing and locating local features of faults; An ultrafast and robust structural damage identification framework enabled by an optimized extreme learning machine;CNN-DBLSTM: A long-term remaining life prediction framework for lithium-ion battery with small number of samples.

4. In the second and third sections, the author introduces the Wegener feature extraction method and the basic principles of object detection, respectively. In addition, the author even introduced the basic principles of convolutional neural networks. However, these methods are mature methods that have been proposed for a long time, and this paper only uses these methods as algorithm frameworks without innovating or improving them. Suggest reducing the length of the basic theory and focusing on elaborating on the work done in this study, such as differences from previous research.

5. The introduction of the data set and experimental preparation section in this paper is relatively detailed. But in the comparative experiment, the persuasiveness is weak because the author did not set any comparison algorithm for comparison, and the comparison angle is single, which cannot intuitively understand the uniqueness and advantages of the work done in this study. In addition, the author did not conduct ablation experiments to demonstrate the advantages of the Wegener method and YOLO object detection algorithm. What is the effect of using other image recognition algorithms combined with the Wegener method? For example, vision transformer.

6. The clarity of the images in this paper needs to be improved, such as Fig. 12, and all font types and sizes in the image need to be consistent.

Comments on the Quality of English Language

The quality of English language still needs improvement.

Author Response

Revised paper of “Electric motor vibration signal classification using Wigner-Ville distribution for fault diagnosis”

Dear Editorial Board

Please find the attached file of the revised paper entitled “Electric motor vibration signal classification using Wigner-Ville distribution for fault diagnosis” which we submitted to “sensors” in December, 2024.  Modifications that have been made to this paper in response to the reviewer’s comments are summarized as follows:

Response to Review #3

  1. This paper proposes a technique for identifying motor fault vibration signals. Firstly, the Wegener method is used to extract vibration signal features, and then the YOLO algorithm is used for fault identification. However, the details of these technologies are not presented in the abstract, which cannot highlight the innovation of this study. Suggest presenting the specific innovative content proposed in this paper in the abstract.

We have added relevant explanations about this study in lines 22-24 of the article.

  1. In the introduction, the author cites a large number of references to verify the multifunctional role of Wegener analysis method in the field of signal processing and the application of YOLO algorithm in the field of computer vision. However, the author did not summarize the above research at the end of the introduction, pointing out the shortcomings that still exist in previous studies and the important contributions proposed in this paper. Suggest adding these contents before introducing the paper layout in the introduction.

We have added relevant explanations about this study in lines 94-110 of the article.

  1. Among the references cited by the author, whether it is Wegener or YOLO method, only a small number of references are relevant to the research objectives of this paper. Suggest the author to enrich the literature content from multiple perspectives, such as: An efficient diagnostic strategy for intermittent faults in electronic circuit systems by enhancing and locating local features of faults; An ultrafast and robust structural damage identification framework enabled by an optimized extreme learning machine;CNN-DBLSTM: A long-term remaining life prediction framework for lithium-ion battery with small number of samples.

In response to this suggestion, the referenced literature in this study has already incorporated the relevant content, specifically in references 16 and 17.

Reference 16 (Tao et al., 2020) discusses the application of wavelet transform and generalized Gaussian density (GGD) modeling for fault detection, providing a robust statistical method to analyze and detect abnormalities in bearing systems.

Reference 17 (Dhandapani et al., 2022) extends this approach by combining generalized Gaussian distribution modeling with multiscale dispersion entropy features, further improving the diagnostic accuracy and classification precision of bearing faults.

  1. In the second and third sections, the author introduces the Wegener feature extraction method and the basic principles of object detection, respectively. In addition, the author even introduced the basic principles of convolutional neural networks. However, these methods are mature methods that have been proposed for a long time, and this paper only uses these methods as algorithm frameworks without innovating or improving them. Suggest reducing the length of the basic theory and focusing on elaborating on the work done in this study, such as differences from previous research.

Thank you for the reminder. The content about CNN has been deleted. And consider adding the principles of YOLOV8 used in this study, in lines 308~312.

  1. The introduction of the data set and experimental preparation section in this paper is relatively detailed. But in the comparative experiment, the persuasiveness is weak because the author did not set any comparison algorithm for comparison, and the comparison angle is single, which cannot intuitively understand the uniqueness and advantages of the work done in this study. In addition, the author did not conduct ablation experiments to demonstrate the advantages of the Wegener method and YOLO object detection algorithm. What is the effect of using other image recognition algorithms combined with the Wegener method? For example, vision transformer.

In line 465 and Table 7, there is a supplement about the method of Pseudo Wigner-Ville Distribution (PWV), and based on the conclusions obtained in the original WVD study, the experiment was started with parameter N. The entire experiment will be improved based on this method to provide a feature extraction method more suitable for fault diagnosis of brushless motors.

  1. The clarity of the images in this paper needs to be improved, such as Fig. 12, and all font types and sizes in the image need to be consistent.

The images clarity has been adjusted, thank you for your reminder.

The paper has been considerably modified. Authors sincerely hope that these modifications have addressed all the points raised by the reviewers and the consistency of the technical content has been improved. Thank you and the referees very much for time and effort needed for the thorough review and the valuable comments on this paper.  I look forward to hearing from you soon.

With best regards,

Jian-Da Wu, Professor

Graduate Institute of Vehicle Engineering

National Changhua University of Education

1 Jin-De Rd., Changhua 500, Taiwan, ROC

Email: jdwu@cc.ncue.edu.tw

Round 2

Reviewer 3 Report

Comments and Suggestions for Authors 1. There is no comparison algorithm set in this paper.   2. The label of the first image in Figure 10 is too small to recognize. Additionally, the format of the following curve graphs should be consistent, such as the position and size of the legend.、   3. Suggest the author to enrich the literature content from multiple perspectives, such as: CNN-DBLSTM: A long-term remaining life prediction framework for lithium-ion battery with small number of samples. Comments on the Quality of English Language

The quality of English still needs improvement.

Author Response

Revised paper of “Electric motor vibration signal classification using Wigner-Ville distribution for fault diagnosis”

Dear Editorial Board

Please find the attached file of the revised paper entitled “Electric motor vibration signal classification using Wigner-Ville distribution for fault diagnosis” which we submitted to “sensors” in December, 2024.  Modifications that have been made to this paper in response to the reviewer’s comments are summarized as follows:

Response to Review #3

  1. There is no comparison algorithm set in this paper.  

In the experiment, both the YOLOv8s and YOLOv8n models were previously utilized for training. The results demonstrated that YOLOv8n achieved superior performance under the given experimental conditions. The YOLOv8s model required higher computational resources, including a more advanced GPU, a larger dataset, or an increased number of training epochs to reach optimal performance. Therefore, YOLOv8n was selected for sample training in this study due to its efficiency and compatibility with the available computational resources.

  1. The label of the first image in Figure 10 is too small to recognize. Additionally, the format of the following curve graphs should be consistent, such as the position and size of the legend.

The image and its dimensions have been adjusted accordingly. Appreciation is extended for the reminder.

  1. Suggest the author to enrich the literature content from multiple perspectives, such as: CNN-DBLSTM: A long-term remaining life prediction framework for lithium-ion battery with small number of samples.

In response to this suggestion, additional diverse references have been incorporated into this study, specifically in references 27 and 28.

Furthermore, the relevant content has been discussed in the article between lines 88 and 91.

The paper has been considerably modified. Authors sincerely hope that these modifications have addressed all the points raised by the reviewers and the consistency of the technical content has been improved. Thank you and the referees very much for time and effort needed for the thorough review and the valuable comments on this paper.  I look forward to hearing from you soon.

With best regards,

Jian-Da Wu, Professor

Graduate Institute of Vehicle Engineering

National Changhua University of Education

1 Jin-De Rd., Changhua 500, Taiwan, ROC

Email: jdwu@cc.ncue.edu.tw
